# Synthesis and Bioevaluation of New Stable Derivatives of Chrysin-8-*C*-Glucoside That Modulate the Antioxidant Keap1/Nrf2/HO-1 Pathway in Human Macrophages

**DOI:** 10.3390/ph17101388

**Published:** 2024-10-17

**Authors:** Javier Ávila-Román, Lirenny Quevedo-Tinoco, Antonio J. Oliveros-Ortiz, Sara García-Gil, Gabriela Rodríguez-García, Virginia Motilva, Mario A. Gómez-Hurtado, Elena Talero

**Affiliations:** 1Department of Pharmacology, Faculty of Pharmacy, University of Seville, 41012 Seville, Spain; sgarcia18@us.es (S.G.-G.); motilva@us.es (V.M.); etalero@us.es (E.T.); 2Instituto de Investigaciones Químico Biológicas, Universidad Michoacana de San Nicolás de Hidalgo, Michoacan 58030, Mexico; 0300768x@umich.mx (L.Q.-T.); antonio.oliveros@umich.mx (A.J.O.-O.); gabriela.rodriguez@umich.mx (G.R.-G.); mario.gomez@umich.mx (M.A.G.-H.)

**Keywords:** chrysin-8-*C*-glucoside derivatives, Nrf2, HO-1, docking, Keap1, dexamethasone, macrophages

## Abstract

**Background/Objectives**: The beneficial effects of the flavonoid chrysin can be reduced by its poor oral bioavailability. It has been shown that chrysin-8-C-glucoside (**1**) has a better absorption capability. The aim of this study was to evaluate the antioxidant and anti-inflammatory activity of this glucoside, as well as the respective hexa-acetate derivative **1a** and the hexa-ethyl carbonate derivative **1b** since the inclusion of moieties in bioactive molecules may increase or modify their biological effects. **Methods**: THP-1 macrophages were used to determine the viability in the presence of chrysin derivatives, and non-cytotoxic concentrations were selected. Subsequently, lipopolysaccharide (LPS)-induced reactive oxygen species (ROS) production and inflammatory mediators were examined. The involvement of chrysin derivatives with the Keap1 and Nrf2 antioxidant system was determined by docking and Western blotting studies. **Results:** Our data demonstrated, for the first time, that pretreatment with the three compounds caused a significant reduction in LPS-induced reactive oxygen species (ROS) production and pro-inflammatory cytokines tumor necrosis factor alpha (TNF-α) and interleukin 1β (IL-1β) levels, as well as in cyclooxygenase 2 (COX-2) expression. The mechanisms underlying these protective effects were related, at least in part, to the competitive molecular interactions of these phenolic compounds with Kelch-like ECH-associated protein 1 (Keap1)–nuclear factor erythroid 2-related factor 2 (Nrf2), which would allow the dissociation of Nrf2 and its translocation into the nucleus and the subsequent up-regulation of hemo-oxygenase 1 (*HO-1*) expression. **Conclusions:** Compared to the 8-*C*-glucoside parent chrysin, compound **1a** exhibited the strongest antioxidant and anti-inflammatory activity. We hypothesized that the incorporation of an acetate group (**1a**) may reduce its polarity and, thus, increase membrane permeability, leading to better pharmacological activity. These findings support the potential use of these phenolic compounds as Nrf2 activators against oxidative-stress-related inflammatory diseases.

## 1. Introduction

Acute inflammation is the immune system’s response to foreign agents, including pathogens, toxic agents, or irradiation. During this process, immune cells, including lymphocytes and macrophages, are activated and release pro-inflammatory and anti-inflammatory mediators, as well as reactive oxygen species (ROS), aimed at eliminating harmful stimuli, repairing injured tissue, and restoring homeostasis. However, non-resolving acute inflammation may progress to persistent chronic inflammation, which has a crucial role in the pathogenesis of many diseases, such as inflammatory bowel disease, rheumatoid arthritis, neurodegenerative diseases, asthma, multiple sclerosis, cardiovascular diseases, and cancer, among others [1]. Accumulating evidence shows that the overproduction of ROS and antioxidant system deficiencies can lead to an oxidative environment, which is also involved in the development of these chronic inflammatory pathologies [2].

Nuclear factor erythroid 2-related factor 2 (Nrf2) is one of the major transcription factors against oxidative stress and inflammation. Under normal conditions, Nrf2 forms a cytoplasmic complex with its negative regulator Kelch-like ECH-associated protein 1 (Keap1), which suppresses the transcriptional activity of Nrf2 through ubiquitination and proteasomal degradation. Nevertheless, under oxidative conditions, Nrf2 dissociates from Keap1 and then translocates to the nucleus and promotes the gene expression of endogenous antioxidant proteins, such as HO-1 [3]. Increased levels of HO-1 mediated by Nrf2 activation have been reported to have anti-inflammatory effects via the suppression of many pro-inflammatory mediators, including cyclooxygenase 2 (COX-2), tumor necrosis factor alpha (TNF-α), and interleukin 1β (IL-1β) levels [4]. Therefore, the search for new Nrf2 activators that suppress excessive oxidative stress and inflammation could be a promising strategy for the treatment of chronic inflammatory diseases.

Flavonoids are a huge family of phenolic compounds with many biological activities, for example, as antioxidant, anti-inflammatory, anticancer, and antimicrobial compounds [5,6]. However, these compounds present a low oral bioavailability, which is enhanced when they are administered in the form of flavonoid glycosides since they possess a better absorption capability [7]. The presence of glucosides is widely reported in the literature, including O- and C-glucosides, which provide variation in chemical properties, highlighting thermodynamics and conformational preferences [8] that are directly related to interactions against biological targets [9]. In this regard, the C-C linkage between an aglycone and its glucoside motif highlights a subtle but important difference in molecular thermodynamics and biological potential compared to the glucosidic C-O linkage [10]. It is good to consider that although *O*-glucoside derivatives have a better absorption capability than their aglycones, the spontaneous hydrolysis of the C-O linkage could occur in the cell, and consequently, the biological potential may be attributed to the aglycone motif. On the contrary, the C-C glucoside linkage is structurally more stable and does not hydrolyze in the gastric and intestinal medium; thereby, their biological potential may be attributed to the overall molecular structure [11].

In particular, chrysin is a flavonoid found in dietary food, such as honey, propolis, and fruit with well-known pharmacological activities such as anticancer, antioxidant, and anti-inflammatory activities [12]. Furthermore, previous studies have evaluated the potential of chrysin as a nutraceutical in diverse experimental models including Parkinson’s disease, Alzheimer’s disease, atopic dermatitis, asthma, ulcerative colitis, and allergic diseases [13,14]. In addition, it has been reported that chrysin O- and C-glucoside derivatives can be found in nature, such as in *Mimosa rosei* B.L.Rob, which is a tree from the Fabaceae family native to the west of Mexico that grows primarily in the seasonally dry tropical biome [15]. Most of the studies reporting the biological properties of these chrysin glucosides evaluated 7-*O*-*β*-D-glucopyranosylchrysin, evidencing antimicrobial and moderated antioxidant effects [16] and diuretic and hypotensive activities [17], as well as immunostimulant potential for cancer treatment [15] and estrogenic activity [18]. However, a previous paper has reported the biological effects of 8-*C*-*β*-D-glucopyranosylchrysin through slightly inducing Nrf2 activation [19].

The pharmacological effects of flavonoids can be related to their OH moieties due to their well-known radical scavenging properties [9]. However, a recent overview highlights the potential of phenol derivatizations to increase their biological potential, which can be achieved by classical and mild methods, including the preparation of carbonyl derivatives, which are able to confer lipophilicity [10]. It is important to note that, currently, no previous studies have reported any structural modifications of chrysin glucosides through derivatization to obtain carbonate or acetate derivatives with the aim of increasing their biological potential. This type of derivatization is simple and reproducible in applications for industrial purposes. Therefore, in this study, two new derivatives of the natural 8-*C*-*β*-D-glucopyranosylchrysin (**1**) from *M. rosei* were synthesized, 8-*C*-(2″,3″,4″,6″-tetraacetyl-*β*-D-glucopyranosyl)-5,7-diacetylchrysin (**1a**) and 8-*C*-(2″,3″,4″,6″-tetra(ethyl carbonate)-*β*-D-glucopyranosyl)-5,7-di(ethyl carbonate)chrysin (**1b**), and their antioxidant and anti-inflammatory activities were compared with those of the parent compound (**1**).

All compounds were characterized by their physical, spectrometric, and spectroscopic data, including melting point, specific rotation, mass spectrometry (MS), infrared (IR), and 1D and 2D Nuclear Magnetic Resonance (NMR). Theoretical physicochemical parameters were estimated with the SwissADME web tool. The purity of all compounds (>96%) was determined by High-Performance Liquid Chromatography (HPLC). In addition, the in chemico ABTS radical scavenging assay was performed. The in vitro assays were carried out using lipopolysaccharide (LPS)-stimulated THP-1 macrophages to evaluate the intracellular ROS production, the pro-inflammatory cytokines TNF-α and IL-1β were determined using ELISA kits, and the expression of COX-2, Nrf2, and HO-1 proteins was evaluated by Western blotting. This cell line was selected to experimentally establish a real approximation to the human macrophage, in addition to being a robust in vitro model for the parameters studied in this work [20,21,22,23]. Finally, a docking analysis using Keap1 (PDB ID: 4ZY3) was conducted. Our results revealed that the three compounds suppressed LPS-induced oxidative stress and the inflammatory response in THP-1 macrophages by reducing ROS levels, as well as TNF-α, IL-1β, and COX-2 production through the activation of the Keap-1/Nrf2/HO-1 pathway, with compound **1a** being the most active.

## 2. Results

### 2.1. Isolation and Chemical Characterization of 8-C-β-D-Glucopyranosylchrysin (***1***) and Preparation of Derivatives ***1a*** and ***1b***

Column chromatography of the methanol extract from the leaves of *M. rosei* provided compound **1** as yellow needles after the crystallization process (MeOH-H_2_O), whose NMR measurements provided resonances of a typical flavonoid glycoside in ^1^H and ^13^C experiments, where the doublet at *δ* 4.69 (*J* = 9.8 Hz) in ^1^H and the signal at *δ* 73.3 in ^13^C suggested a *C*-glucosidic linkage. Two-dimensional NMR experiments supported the ^1^H and ^13^C spectral assignations. The physical and spectroscopic data for the natural compound were consistent with 8-*C*-*β*-D-glucopyranosylchrysin (**1**) [24]. The acetylation of **1** with acetic anhydride was achieved (see experimental section) to yield a white amorphous solid whose HRMS (ESI^+^) showed *m/z* 669.1814 [M + H]^+^ (calcd for C_33_H_32_O_15_ + H^+^, 669.1819). Its ^1^H NMR data evidenced a resonance pattern from the flavone portion at *δ* 7.84–6.65 and the expected *C*-glucose signal pattern in the range of *δ* 5.74–3.82, as well as six singlet signals in the gap of *δ* 2.51–1.82. The ^13^C NMR spectrum revealed the integrity of the glucoside flavonoid skeleton, together with six carbonyl signals from the acetate moiety in the range of *δ* 170.4–167.8 and respective CH_3_ resonances in the range of *δ* 29.6–20.0, thus 8-*C*-(2″,3″,4″,6″-tetraacetyl-*β*-D-glucopyranosyl)-5,7-diacetylchrysin (**1a**). The derivatization of **1** with ethyl chloroformate provided a white amorphous solid (see experimental section). The HRMS of this derivative (ESI^+^) showed *m/z* 849.2449 [M + H]^+^ (calcd for C_39_H_44_O_21_ + H^+^, 849.2459). The ^1^H NMR spectrum showed the expected ethyl moiety resonances in the range of *δ* 4.40–3.75 (CH_2_) and 1.46–1.20 (CH_3_), as well as a resonance pattern from the C-glucose moiety in the range of *δ* 5.32–4.07. ^13^C NMR revealed the presence of six signals in the range of *δ* 154.8–151.2, which were related to carbonate moieties, as well as signals from flavone (*δ* 177.4–108.3) and glucose portions (*δ* 77.0–65.1), thus revealing the formation of 8-*C*-(2″,3″,4″,6″-tetra(ethyl carbonate)-*β*-D-glucopyranosyl)-5,7-di(ethyl carbonate) chrysin (**1b**). The formulas of compounds **1**, **1a,** and **1b** are depicted in Figure 1.

### 2.2. Antioxidant Scavenging Activity

To evaluate the antioxidant capacity of compounds **1**, **1a**, and **1b,** we examined their scavenging activity by using the ABTS assay. This method is often reported as EC_50_, which is defined as the half maximal effective concentration of the antioxidant required to reduce the initial ABTS absorbance by 50%. Among these three phenolics, compound **1** exhibited the most notable antioxidant capacity, exhibiting an EC_50_ of 366.68 µM (or 15.18 Trolox equivalents). The phenolic compounds **1a** and **1b** hardly showed antioxidant activity as scavengers (Appendix A).

### 2.3. Intracellular ROS Measurements

The ROS levels in THP-1 macrophages were determined using DCF-DA (Figure 2). First, the cytotoxicity of the compounds was evaluated with the MTT assay. The results evidenced that **1** and **1b** did not show cytotoxicity at 100 μM, but **1a** affected cell viability by 30% from 50 μM (Appendix A). Therefore, to rule out cytotoxic effects, compounds **1** and **1b** were tested at 50 and 100 µM and compound **1a** at 5 and 10 μM. LPS induced a significant increase in ROS production compared to the Control group, which showed 61.6% of ROS levels. Dexamethasone (Dex), used as a positive reference compound, reduced ROS production in a similar way to the Control group. Treatment with phenolic **1** showed a notable decrease in ROS production, being significative at the higher tested concentration (100 µM), which achieved similar levels to those of the Control group. Compound **1a** statistically reduced the ROS production, reaching levels lower than those of the Control group at concentrations of 5 µM (40.9%) and 10 µM (34.9%), being the most active phenolic tested. Pretreatment with compound **1b** also resulted in a significant decrease in ROS levels at concentrations 50 and 100 µM, showing 46.7% and 21.6% of ROS production, respectively, reaching lower levels than in the Control group.

### 2.4. Pro-Inflammatory Cytokine Production

The anti-inflammatory activity of **1** obtained from *M. rosei* B.L.Rob and its derivatives **1a** and **1b** was tested by measuring TNF-α and IL-1β levels in LPS-stimulated THP-1 macrophages. Non-cytotoxic concentrations were used to carry out this assessment; compounds **1** and **1b** were tested at 10, 50 and 100 µM and the compound **1a** was tested at 1, 5 and 10 μM. As shown in Figure 3, LPS significantly increased TNF-α and IL-1β production in THP-1 cells in relation to unstimulated Control cells (*p* < 0.001). As expected, Dex, used as an anti-inflammatory reference drug, induced a marked inhibition in LPS-induced TNF-α and IL-1β production (Figure 3A,B, respectively). Pretreatment of cells with compound **1** substantially reduced TNF-α levels with the higher concentration (*p* < 0.001) and IL-6 levels with the concentrations of 50 and 100 µM (*p* < 0.05 and *p* < 0.01, respectively). As regards compound **1a**, it significantly inhibited TNF-α and IL-6 levels at the concentration of 10 µM (*p* < 0.01 and *p* < 0.05, respectively). In addition, the pretreatment with compound **1b** induced a significant reduction in these cytokine levels with the concentrations of 50 and 100 µM (TNF-α, *p* < 0.01 and IL-6, *p* < 0.05).

### 2.5. Anti-Inflammatory and Antioxidant Protein Expression

To explore the potential anti-inflammatory mechanism of action of phenolics **1**, **1a**, and **1b**, the expression of the pro-inflammatory enzyme COX-2 was measured by Western blotting. THP-1 cells were pretreated with different concentrations of flavonoids **1**, **1b** (10, 50, and 100 μM), **1a** (1, 5, and 10 μM), and Dex (1 μM) for 1 h and then stimulated with LPS (1 μg/mL). As presented in Figure 4A, LPS stimulation caused a marked increase in COX-2 protein levels in THP-1 macrophages compared to Control cells (*p* < 0.001) (Figure 4B). The positive reference compound Dex significantly reduced COX-2 levels in the presence of LPS (*p* < 0.001). Pretreatment with phenolics **1** and **1b** resulted in a statistical decrease in COX-2 levels at 100 μM (*p* < 0.05 and *p* < 0.01, respectively), and compound **1a** was the most effective in reducing COX-2 expression at 5 and 10 μM (*p* < 0.05 and *p* < 0.01, respectively) (Figure 4B).

To identify whether phenolics **1**, **1a**, and **1b** could attenuate LPS-induced oxidative stress through the regulation of the Nrf2-dependent antioxidant signaling pathway, Nrf2 and HO-1 protein expressions were measured by Western blotting. As presented in Figure 4A, LPS stimulation induced a statistically significant reduction in the expression of Nrf2 compared to Control cells (*p* < 0.01) (Figure 4C). The positive reference Dex showed a significant up-regulation of Nrf2 levels (*p* < 0.05) in comparison with the LPS group. Similarly, the three phenolics were significantly able to up-regulate Nrf2 protein levels at all assayed concentrations in comparison with the LPS group (Figure 4C), showing a recovery comparable to Control levels. These results were correlated with a statistical increase in its target gene *HO-1* with phenolic **1** at 50 and 100 µM (*p* < 0.05), **1a** at 10 µM (*p* < 0.05), and **1b** at 10, 50, and 100 µM (*p* < 0.05) (Figure 4D).

### 2.6. Docking Study

The validation of the protocol was achieved by using the native ligand N,N′-[2-(2-oxopropyl)naphthalene-1,4-diyl]bis(4-ethoxybenzenesulfonamide). Thus, the native ligand was re-docked within the Kelch domain, showing reliability and reproducibility for this study (Table 1). The Kelch domain from Keap1 is constituted by a six-bladed *β*-propeller structure [25]. The docked conformation of the native ligand in the active site coordinates of Keap1 (−51.176, −3.868, −7.609 for x, y, z, respectively) was highly similar to that of the native co-crystallized ligand (Appendix A), suggesting that subsequent protocols using **1**, **1a**, and **1b** are valid. The lowest binding affinity was −9.3 kcal/mol for the best-docked pose, whose root mean square deviation (RMSD) value was 0.811 Å. Compounds **1**, **1a**, and **1b** were docked to Keap1 target protein 4ZY3, and the binding affinities are reported in Table 1. The above results revealed that compound **1** has a higher binding affinity than **1a** and **1b**.

Compound **1** had a −8.9 kcal/mol binding affinity, where the amino acids from the protein target involved in the hydrogen bonding formation include Ser363, Asn382, Asn414, Arg415, Gln530, and Ser602, while pi–pi stacked and pi–alkyl interactions involve Tyr572 and Ala556, respectively (Figure 5A). Compound **1a** revealed two poses (**1a**P1 and **1a**P2) at a −8.2 kcal/mol binding energy (Figure 5B,C); however, one of them (**1a**P1) showed both favorable and unfavorable receptor–receptor interactions (Table 1) and was consequently discarded. The second pose (**1a**P2, Figure 5C) revealed conventional hydrogen bond interactions with Gly364, Arg380, Asn382, and Arg415. Non-classical hydrogen bond interactions for Keap1–**1a** include Ser363, Gly603, and Gly462, while pi–pi stacked, pi–cation, and pi–alkyl interactions involve Tyr525, Arg483, and Ala556, respectively (Table 1). Compound **1b** had −7.9 kcal/mol binding energy values. This compound showed three poses at the best binding energy (**1b**P1, **1b**P2, **1b**P3), as depicted in Figure 5D,E,F, where recurrent interactions were found and include Arg380, Asn382, Arg415, Arg483, Gly509, and Ala556 (Table 1). Other intermolecular interactions are listed in Table 1. Conformer **1b**P1 showed an additional pi–alkyl interaction with Tyr334. Conformer **1b**P2 provided the highest number of Keap1–**1b** interactions, where an additional hydrogen bond with Gln530 and carbon–hydrogen bond interactions with Gly462 were observed (Figure 5E). Moreover, weaker molecular interactions were found, including pi–sigma with Tyr525 and pi–alkyl with Tyr572 and Phe577. Finally, Conformer **1b**P3 (Figure 5F) showed an additional interaction with Ser602 and non-classic hydrogen bonds with Gly462.

### 2.7. In Silico Analysis of the Physicochemical Properties of ***1*** and ***1a***

The theoretical lipophilicity parameters for compounds **1** and **1a** were estimated using the SwissADME web tool [26]. The prediction for compound **1** estimated iLOGP 1.26, XLOGP3 0.41, WLOGP −0.23, MLOGP −2.02, and SILICOS-IT 0.33. The prediction for compound **1a** estimated iLOGP 3.77, XLOGP3 1.93, WLOGP 2.71, MLOGP 0.74, and SILICOS-IT 4.02. Additional theoretical physicochemical parameters estimated with the SwissADME web tool can be found in the Appendix A.

## 3. Discussion

Chrysin is a flavonoid with a broad spectrum of biological activities [12]. However, its poor oral bioavailability may reduce its efficacy; to overcome this limitation, this flavonoid can be administered in the form of glycosides, as they possess a better absorption capacity. Moreover, the synthetic derivatization of chrysin glycosides could be a promising strategy, as it may increase the pharmacological efficacy. Several studies aiming to improve the anti-inflammatory potential of the chrysin core focused on the derivatization of the 8-*C*-position with interesting results, particularly when a pyridyl or hydroxyethyl moieties were linked [27,28]. In nature, the presence of chrysin-8-*C*-glucoside (**1**) is reported [24], which represents a strategical compound to study the biological improvement of the chrysin skeleton for anti-inflammatory purposes, considering the advantage that the glucoside functional group is expected to be a biocompatible motif due to its biochemical role in cells [29]. In the present work, the antioxidant and anti-inflammatory potential of the natural chrysin-8-*C*-glucoside **1**, the respective acetate derivative **1a**, and the ethyl carbonate derivative **1b** was evaluated, since the inclusion of moieties in bioactive molecules may increase or modify their biological potential. Our results showed, for the first time, the anti-inflammatory effects of the two chrysin-8-*C*-glucoside derivatives in LPS-induced THP-1 macrophages via the up-regulation of the Nrf2/HO-1 pathway. This activity may be related to the overall 8-*C*-glucoside molecular skeleton since the stability of the C-C glucosidic linkage is warranted, thus hypothesizing a particular biological behavior when compared with the 8-*O*-glucoside or aglycone analogues [11].

First, the isolation of **1** from *M. rosei* was achieved after purification by chromatography. The chemical characterization of **1** by NMR measurements provided resonances of a typical flavonoid glycoside in ^1^H and ^13^C experiments, where the doublet at *δ* 4.69 (*J* = 9.8 Hz) in ^1^H and the signal at *δ* 73.3 in ^13^C suggested a *C*-glucosidic linkage. Two-dimensional NMR experiments supported the ^1^H and ^13^C spectral assignations. The physical and spectroscopic data for the natural compound were consistent with previous reports [24], thus ensuring the presence of 8-*C*-*β*-*D*-glucopyranosylchrysin (**1**). It is worth mentioning that *M. rosei* was recently evaluated by our research group, reporting detailed molecular structure studies of several chrysin glucoside derivatives, including chemical reactivity to guarantee per-acetylation derivative preparation [15]. Thus, the per-acetylation of compound **1** yielded compound **1a** as a white solid whose ^1^H NMR spectrum revealed solved signal patterns in the range of *δ* 5.73–3.83 assigned to the glucose moiety and six new singlet signals in the range of *δ* 2.50–2.48 related to acetate residues. The ^13^C NMR spectrum showed resonances of six additional acetate moieties in the range of *δ* 170.4–167.8, which were related to sp^2^-hybridized carbons, as well as six new signals in the range of *δ* 29.6–20.0 related to methyl from the acetate portions. Regarding compound **1b**, it was prepared by using ethyl chloroformate as an esterification agent and the above mild methodology [15] to provide a white solid whose ^1^H NMR spectrum showed overlapped signals in the range of *δ* 4.40–3.75, as well as in the range of *δ* 1.46–1.20, thus, the expected derivatization. The ^13^C NMR spectrum revealed six new carbonyl signals in the range of *δ* 154.7–151.3 as key signals to ensure the obtention of the percarbonate compound.

An imbalance between ROS production and the antioxidant defense system has been reported to result in excessive oxidative stress that further promotes the inflammatory process and leads to the development of chronic diseases [2]. Therefore, the search for new compounds that regulate the cellular redox status may provide a therapeutic strategy against oxidative-stress-related inflammatory diseases. Chrysin has previously been reported to exhibit a marked free radical scavenging activity according to our findings [30]. However, the chrysin derivatives **1a** and **1b** studied herein hardly showed a free radical scavenging activity, which could be related to the motifs linked to the flavonoid skeleton, which could be considered as protecting groups of the phenolic OH group, leading to a remarkable reduction in the antioxidant capacity when compared with the typical values found for flavonoid compounds [31]. Furthermore, the antioxidant activity of chrysin has previously been reported in different cellular models [32,33]. Based on these findings, the antioxidant effect of the phenolic compounds was evaluated in LPS-stimulated THP-1 macrophages as an in vitro model of oxidative stress and inflammation by using DCF-DA. Our data demonstrated that the three compounds caused a marked reduction in LPS-induced ROS production, compound **1a** being the most active, since at the lower concentrations tested it displayed similar results to compounds **1** and **1b**. During the inflammatory process, LPS-activated macrophages stimulate the progression of inflammation through the overproduction of pro-inflammatory cytokines, including TNF-α and IL-1β. These mediators play a crucial role in the pathogenesis of a variety of inflammatory diseases, such as inflammatory bowel diseases, psoriasis, rheumatoid arthritis, and cancer, among others [34]. For this reason, some inhibitors for these cytokines have been used as an effective strategy for the treatment of these pathologies [35,36]. Consistent with these findings, chrysin has been shown to have anti-inflammatory activity by reducing several pro-inflammatory cytokines, such as TNF-α and IL-1β, among other pro-inflammatory parameters in both in vitro and in vivo models of oxidative-stress-related inflammatory diseases [37,38,39,40,41,42,43,44,45,46,47,48,49,50]. In particular, the present work is the first reporting the biological activity of chrysin glucoside derivatives influencing TNF-α and IL-1β cytokine levels.

A growing number of studies have revealed that the long-term up-regulation of the inducible enzyme COX-2 leads to an excessive synthesis of pro-inflammatory prostaglandins such as PGE_2_, which exacerbates the chronic inflammatory response [51]. Our data are in agreement with other in vitro studies reporting that chrysin or derivatives inhibited COX-2 expression in RAW264.7 macrophages [52] and human osteoarthritis chondrocytes [53].

The multifunctional regulator Nrf2 controls the expression of a variety of cytoprotective and antioxidant genes, such as HO-1, which has been shown to down-regulate many pro-inflammatory mediators [4]. Chrysin has been reported to have potent antioxidant and anti-inflammatory activity by inducing the Nrf2/HO-1 axis in different in vitro and in vivo inflammation models [32,54]. To our knowledge, only one study has demonstrated that chrysin-8-*C*-glucoside induces a slight Nrf2 activation [19], and no studies have evaluated the effects of acetate or ethyl carbonate derivatives on the Nrf2/HO-1 pathway. Our data propose that the up-regulation of the Nrf2/HO-1 pathway could prevent the increase in ROS levels and the pro-inflammatory mediators TNF-α, IL-1β, and COX-2, thus attenuating the injury produced by oxidative stress and inflammation. This activity may be related to the overall 8-*C*-glucoside molecular skeleton since the stability of the C-C glucosidic linkage is warranted, thus hypothesizing a particular biological behavior when compared with the 8-*O*-glucoside or aglycone analogues [11].

According to the up-regulation of the Nrf2/HO-1 signaling pathway shown by **1**, **1a**, and **1b**, we further researched the underlying mechanisms by studying the potential interactions of these glucosidic phenols with the natural inhibitor of Nrf2, Kelch-like ECH-associated protein 1 (Keap1), based on a previous study in which a flavonoid chalcone-type interacts with Keap-1 [55], by using a docking study. These docking analyses assume the integrity of flavonoid derivatives **1a** and **1b** after entering the cell, since the glucose moieties are linked by covalent C-C bonds to the chrysin portion. Our chrysin derivatives interacted in the Kelch domain from Keap1 [25] through stronger intermolecular interactions such as hydrogen bonds, which are favored by the residues from polar amino acids (e.g., Arg, Asn, Gly, and Tyr), or through mild interactions that involve Pi-electrons from aromatic motifs, such as those from tyrosine. In particular, recurrent molecular interactions with Keap1 occur between A and B rings with Ala556, Arg415, and Asn382 amino acids (Table 1), as illustrated in Figure 6 when **1** interacts with Keap1, thus suggesting those portions of flavonoids as a key functionality for enzyme–substrate interactions [56].

This study highlights the important role of the glucosidic motif, since pyranic oxygen from **1** or carbonyl groups from **1a** and **1b** promote additional anchoring sites besides the expected improvement in pharmacokinetics [29] and thus are considered as strategic moieties. However, two conformers for **1a** and three conformers for **1b** were evidenced to be suitable poses for enzyme–substrate interactions, suggesting conformational freedom during enzyme–substrate interactions, thus showing a dissimilar affinity of **1a** and **1b** compared to compound **1**. The above conformational freedom is consistent with the results of the calculation of the conformational distribution, where the compound **1a** model provided 157 conformers, and the derivative **1b** showed 591 conformers in the 0–5 kcal/mol gap, which is deeply related to the rotational freedom of the acetate and carbonate moieties, respectively. It is worth mentioning that the conformer distribution of compound **1** provided 43 conformers in the same energy window, revealing increased conformational restrictions compared to compounds **1a** and **1b**, which could be related to the docking results. Interestingly, the docking results were consistently related to the experimental results described herein. In this regard, the literature suggested that Arg415, Arg483, Ser508, Arg380, Ser363, and Asn382 from Keap1 support the main intermolecular interactions to promote the Keap1–Nrf2 system [57,58]. Our results evidenced that **1a**P2 promotes closely similar interactions in Keap1–**1a**P2 docking compared to the Keap1–Nrf2 interaction, with the exception of Ser508, whose interaction with **1a** was missed in our computational approximation. Furthermore, the conformers **1b**P1, **1bP2,** and **1b**P3 also suggested relevant competition for that strategic target from Keap1, where each conformer revealed interactions with Arg415, Arg483, Arg380, and Asn382. Curiously, compound **1** displayed interactions only with Arg 415 and Asn380, despite this flavonoid providing the best energy interaction in the docking study, thus suggesting that these amino acids could be considered as key motifs from Keap1 to provide adequate competition against the Keap1–Nrf2 interaction. Taking together these results, we hypothesized that the compounds **1**, **1a**, and **1b** could act as Nrf2 activators through the competitive molecular interactions of these phenolics with Keap1–Nrf2. These interactions may lead to the dissociation of Nrf2 from the cytoplasmic complex with Keap1, which would allow its activation and translocation into the nucleus, leading to the up-regulation of HO-1 expression and a reduction in the inflammatory process (Figure 7). Moreover, this proposed mechanism may explain the up-regulation of *HO-1* expression observed with the phenolic compounds despite them only being capable of recovering an *Nrf2* expression level similar to that of the Control. Additionally, since all derivatives studied herein contain a glucose moiety, it may support a higher cell permeability of these compounds [59,60]. The incorporation of mild apolar motifs such as the acetate groups in compound **1a** provides higher lipophilicity according to bioinformatic approximations determined herein (Appendix A) by using SwissADME web tool [26], thus deriving a potential increased membrane permeability [61] of all the tested phenolics and, subsequently, greater antioxidant and anti-inflammatory activity.

## 4. Materials and Methods

### 4.1. General Experimental Procedures

Melting points (uncorrected) were recorded on a Fisher-Johns apparatus (Thermo Fisher Scientific Inc., Waltham, MA, USA). MS was carried out on a Varian Saturn 2000 spectrometer (Agilent Technologies, Inc., Santa Clara, CA, USA). HRMS spectra were acquired on a Bruker MicroTOF-II spectrometer (Bruker, Corp., Billerica, MA, USA). One-dimensional and two-dimensional NMR spectra were measured at 400 MHz for ^1^H and 100 MHz for ^13^C on the Varian Mercury 400 spectrometer (Agilent Technologies, Inc., Santa Clara, CA, USA) from dimethyl sulfoxide (DMSO)-d6 or CDCl_3_ solutions. Chemical shift (δ) values are expressed in parts per million relative to TMS, and coupling constants (J) are reported in Hz.

Purity analysis of samples was quantified using an HPLC 1260 Infinity II LC System (Agilent Technologies, Inc., Santa Clara, CA, USA) equipped with a photodiode array detector (Agilent model G4212B, 320 and 280 nm) and a 4.6 × 250 mm XSelect^®^ HSSC18 (5 μm) (Waters, Milford, MA, USA). An isocratic mobile phase (at 1 mL/min) consisting of 50% and 50% aqueous acetic acid (1%)–acetonitrile for compound **1** was used, while a 50% and 50% aqueous acetic acid (0.5%)–acetonitrile mixture was employed for compounds **1a** and **1b**. Water and acetonitrile were of HPLC grade from Merck. The mobile phases were degassed for 15 min in an ultrasonic bath before use. All tested compounds possessed a purity of over 96% (Appendix A).

### 4.2. Extraction and Isolation of 8-C-β-D-Glucopyranosylchrysin (***1***)

The isolation of 8-*C*-*β*-D-glucopyranosylchrysin (**1**) was achieved by successive column chromatography from the methanol extract from leaves of *M. rosei* B.L.Rob according to the previously described methodology [15]. The isolated product was analyzed by spectroscopic and spectrometric means, and then, it was compared with an authentic sample.

### 4.3. General Procedure of Acetylation and Ethyl Carbonate Derivatizations

A batch of **1** (10 mg) was dissolved in pyridine (0.5 mL), and 1 mL of Ac_2_O was added. The mixture was stirred and heated in a steam bath for 4 h. After, the reaction mixture was poured over wet ice and extracted with ethyl acetate. The organic layer was washed with aqueous HCl (10%), distilled water, saturated aqueous NaHCO_3_ solution, and water, dried over Na_2_SO_4_, filtered, and evaporated. The process yielded **1a** (12.6 mg, 80%).

For ethyl carbonate derivatizations, the above procedure was conducted but using ethyl chloroformate instead of Ac_2_O and using 5% HCl aqueous solution for the extraction process. The reaction yielded **1b** (11.2 mg, 56%). The methodology for the obtention and chemical characterization of **1**, **1a**, and **1b** is depicted in Figure 8.

***8-C-β-**D-Glucopyranosylchrysin****** (*****1*).*** Yellow needles (MeOH-H_2_O), m.p. 242–245 °C, IR *ν*_max_ (cm^−1^): 3485, 3384, 1646, 1613, 1062, and 843. IR, ^1^H, and ^13^C NMR (Appendix A) data were concordant with the literature [24] and supported by 2D NMR (Appendix A).

***8-C-(2″,3″,4″,6″-Tetraacetyl-β-**D-glucopyranosyl)-5,7-diacetylchrysin (*1a*).*** White amorphous solid, m.p. 124–126 °C. IR *ν*_max_ (cm^−1^): 2925, 1747, 1646, 1165, and 1033. ^1^H NMR (400 MHz, CDCl_3_) *δ* 7.84 (2H, dd, *J* = 8.0, 1.6, H-2′, H-6′), 7.52 (3H, m, H-3′, H-4′, H-5′), 7.34 (1H, s, H-6), 6.65 (1H, s, H-3), 5.74 (1H, t, *J* = 9.6, H-2″), 5.33 (1H, t, *J* = 9.6, H-3′), 5.18 (1H, t, *J* = 9.6, H-4″), 4.86 (1H, d, *J* = 9.6, H-1″), 4.42 (1H, dd, *J* = 12.0, 4.0, H-6a″), 4.00 (1H, d, *J* = 12.0, H-6b″), 3.82 (1H, ddd, *J =* 9.6, 4.0, 1.6, H-5″), 2.51 (3H, s, Ac-7), 2.49 (3H, s, Ac-5), and 2.07–1.82 (Ac from sugar); ^13^C NMR (100 MHz, CDCl_3_), *δ* 176.1 (C-4), 170.4 (Ac-5), 170.2 (Ac-7), 169.5–167.8 (Ac—sugar), 162.5 (C-2), 157.3 (C-7), 153.2 (C-5), 148.6 (C-9), 131.8 (C-4′), 130.7 (C-1′), 129.0 (C-3′, C-5′), 126.1 (C-2′, C-6′), 118.9 (C-8), 111.7 (C-3), 108.6 (C-6), 89.4 (C-5″), 76.4 (C-3″), 72.3 (C-1″), 70.4 (C-2″), 68.0 (C-4″), 60.3 (C-6″), 29.6 (Ac-5), 29.6 (Ac-7), and 21.0–20.0 (Ac—sugar). The 1D and 2D NMR spectra are depicted in Appendix A.

***8-C-(2″,3″,4″,6″-Tetra(ethyl carbonate)-β-**D-glucopyranosyl)-5,7-di(ethyl carbonate) chrysin (*1b*).*** White amorphous solid, m.p. 75–77 °C. IR *ν*_max_ (cm^−1^): 2929, 1775, 1649, 1204, and 1018. ^1^H NMR (CDCl_3_, 400 MHz) *δ* 7.87 (2H, dd, *J* = 7.6, 1.6, H-2′, H-6′), 7.52 (3H, m, H-3′, H-4′, H-5′), 7.06 (1H, d, *J* = 2.4, H-8), 6.78 (1H, d, *J* = 2.4, H-6), 6.67 (1H, s, H-3), 5.32 (1H, d, *J* = 7.6, H-1″), 5.24 (1H, t, *J* = 9.2, H-4″), 5.16 (1H, dd, *J* = 9.2, 7.6, H-2″), 5.03 (1H, t, *J* = 9.2, H-3″), 4.33 (1H, dd, *J* = 12.0, 2.8, H-6″), 4.07 (1H, ddd, *J* = 9.2, 6.0, 2.8, H-5″), 4.40–3.75 (m, CH_2_ protons from ethyl carbonate), and 1.46–1.20 (m, CH_3_ protons from ethyl carbonate). ^13^C NMR (100 MHz, CDCl_3_) *δ* 177.4 (C-4), 164.3 (C-2, C-7), 162.5 (C-2), 162.3 (C-5), 157.2 (C-9), 154.8, 154.3, 153.8, 153.3, 152.6, 151.2 (C=O), 131.6 (C-4′), 131.5 (C-1′), 129.3 (C-3′, C-5′), 126.4 (C-2′, C-6′), 110.6 (C-10), 110.5 (C-3), 108.4 (C-6), 108.3 (C-8), 77.0 (C-5″), 76.7 (C-3″), 74.6 (C-1″) 73.7 (C-2″), 71.8 (C-4″), 65.1 (C-6″), 65.1–64.5 (CH_2_ carbons from ethyl carbonate), and 14.2–14.0 (CH_3_ carbons from ethyl carbonate). The 1D and 2D NMR spectra are depicted in Appendix A.

### 4.4. Cell Culture

THP-1 is a human leukemia monocytic cell line that was purchased from ATCC (USA). The cells were grown in RPMI 1640 medium (GIBCO^®^, Life Technologies, NY, USA) supplemented with 10% heat-inactivated fetal bovine serum, at 37 °C in a 5% CO_2_ incubator. In addition, penicillin (100 U/mL) and streptomycin (100 mg/mL) (PAA^®^, Pasching, Austria) were added to the culture medium.

### 4.5. Cytotoxicity Assay

The cytotoxicity of compounds was determined in THP-1 macrophages by the 3-(4,5-dimethylthiazol-2-yl)-2,5-diphenyltetrazolium bromide (MTT, Calbiochem, Darmstadt, Germany) assay [62]. THP-1 cells were seeded in 96-well plates (100 μL/well) at a density 10^4^ cells/mL and differentiated into macrophages in the presence of 8 nM phorbol 12-myristate 13-acetate (PMA, Sigma-Aldrich Química, S.A., Madrid, Spain) at 37 °C for 72 h in a humidified atmosphere of 5% CO_2_. After that, the medium was discarded, and the cells were washed twice with phosphate saline buffer (PBS, 4 °C) and then incubated for 48 h with the compounds at different concentrations (0, 6.25, 12.5, 25, 50, and 100 µM), which were freshly prepared in culture medium. Control cells were incubated in culture medium with a non-toxic DMSO concentration (0.5% *v/v*). After washing the cells with PBS, 100 μL of MTT (0.25 mg/mL) was added into the wells and kept for 4 h. The formazan crystals were dissolved using 200 μL of DMSO and 25 μL of 0.1 M glycine buffer pH 10.5. Afterwards, the absorbance was determined on a microplate reader Multiskan EX at a wavelength of 550 nm (Labsystems, Thermo Scientific, Waltham, MA, USA). The experiments were carried out in quadruplicate, and the 50% inhibitory concentration (IC_50_) was calculated.

### 4.6. ABTS Radical Scavenging Assay

The 2,2′-azino-bis(3-ethylbenzothiazoline-6-sulphonic acid) (ABTS) colorimetric method [63] was carried out to test the antioxidant capacity. The ABTS·+ radical was obtained by mixing 3.84 mg/mL of ABTS (Applichem, Darmstadt, the Netherlands) and 6.6 mg/mL of potassium persulfate (Sigma-Aldrich Química, SA, Spain). This solution was macerated at 4 °C in dark conditions for 24 h. Phenolics **1**, **1a**, and **1b** were pipetted onto 96-well plates at a concentration range of 0–200 μM, and a solution of ABTS·+ (equivalent to an absorbance of 0.7 ± 0.02) was added. Next, the absorbance was monitored at 734 nm for 7 min with a microplate reader (Synergy-HT-multimode, BioTek instruments, Thermo Scientific, Waltham, MA, USA). Trolox was used as a reference compound at the same concentration range. The effective concentration 50% (EC_50_) was determined, and data are expressed as the Trolox Equivalent Antioxidant Capacity (TEAC) (EC_50_ sample/EC_50_ Trolox).

### 4.7. Intracellular ROS Production

The production of ROS in THP-1 macrophages was determined by using the DCF-DA assay (abcam, Cambridge, UK). Briefly, 10^4^ cells/well were seeded in 96-well black plates (100 μL/well) in the presence of 8 nM PMA (37 °C, 72 h). Next, the medium was removed, the cells were washed (PBS, 4 °C), and then a pretreatment with **1** and **1b** (0, 50, and 100 µM), **1a** (0, 5, and 10 µM), and Dex as the control (1µM) was performed for 1 h. Then, the stimulation process was activated by using *E. coli* LPS (1 µg/mL) for 24 h. Control cells, both unstimulated and LPS-stimulated, were incubated in cultured medium with a non-toxic DMSO concentration (0.5% *v/v*). Afterwards, the supernatants were removed, the cells were washed (PBS, 4 °C), and subsequently, 20 µM DCFDA (100 µL/well) was added and incubated for 45 min. Fluorescence was measured on a fluorescence plate reader (Sinergy HT, Biotek^®^, Bad Friedrichshall, Germany) with an excitation (485 nm) and an emission wavelength (535 nm) following the instructions of the manufacturer.

### 4.8. Determination of Pro-Inflammatory Cytokine Levels

The assessment of the pro-inflammatory cytokine levels was carried out by using THP-1 macrophages. THP-1 cells were seeded at a density of 10^4^ cells/well using 96-well plates (100 μL/well) in the presence of 8 nM PMA, as described above (37 °C, 72 h). Next, the medium was discarded, and the cells were washed with ice-cold PBS. Then, THP-1 cells were pretreated for 1 h with the phenolic compounds **1** and **1b** (0, 10, 50, and 100 µM), **1a** (0, 1, 5, and 10 µM), and Dex as the positive control (1 µM) and then stimulated for an additional 24 h with LPS (1 µg/mL). Control cells, both unstimulated and LPS-stimulated, were incubated in cultured medium with a non-toxic DMSO concentration (0.5% *v/v*). Next, the supernatants were kept at −80 °C until the detection of the cytokines TNF-α and IL-1β using ELISA kits (Diaclone GEN-PROBE, Besançon Cedex, France), following the manufacturer’s instructions. The absorbance was monitored with a microplate reader (Labsystems Multiskan EX, Thermo Scientific, USA) at 450 nm.

### 4.9. Protein Expression Measurement

The expression of proteins was measured by Western blotting in THP-1 cells. Briefly, for differentiation into macrophages, the cells (8 × 10^5^ cells/well) were incubated with 8 nM PMA in 6-well plates (2 mL/well) (37 °C, 72 h). Afterward, the medium was discarded, and the cells were washed twice with ice-cold PBS. Next, the cells were pretreated for 1 h with the phenolic compounds **1** and **1b** (0, 10, 50, and 100 µM), **1a** (0, 1, 5, and 10 µM), and Dex as a positive control (1 µM) and then stimulated for an additional 24 h with LPS (1 µg/mL). Control cells, both unstimulated and LPS-stimulated, were incubated in cultured medium with a non-toxic DMSO concentration (0.5% *v/v*). After that, total protein was extracted from cells lysed with ice-cold buffer containing a protease inhibitor cocktail (0.01 mg/mL leupeptin, 0.01 mg/mL pepstatin, 0.01 mg/mL aprotinin) and 50 mM Tris-HCl pH 7.5, 8 mM MgCl_2_, 5 mM ethylene glycol-bis(2-aminoethylether)-N,N,N′,N′-tetraacetic acid, 0.5 mM EDTA, 1 mM phenylmethylsulfonyl fluoride, and 250 mM NaCl and incubated on ice for 30 min. Next, cell homogenates were centrifugated for 5 min at 12,000× *g* and 4 °C. Proteins were quantified using the Bradford method, and equal amounts of proteins (50 µg) were loaded onto 10% sodium dodecyl sulfate–polyacrylamide gel electrophoresis (SDS-PAGE). The proteins separated by size were transferred onto a nitrocellulose membrane and incubated with the following primary antibodies: rabbit anti-COX-2 (1:3000; Cayman Chemical, Ann Arbor, MI, USA), rabbit anti-Nrf2 (1:1000), and rabbit anti-HO-1 (1:1000) (Cell Signaling, Danvers, MA, USA). Membranes were washed three times for 10 min and incubated with the secondary horseradish peroxidase-linked anti-rabbit antibody (Pierce Chemical, Rockford, IL, USA). A chemiluminescence light-detecting kit (Super-Signal West Pico Chemiluminescent Substrate, Pierce, IL, USA) was used to visualize protein bands. The proteins were quantified using the Image J software 1.53e (National Institute of Mental Health, Bethesda, MD, USA). β-actin was used as a loading control to normalize the band intensities of the target proteins.

### 4.10. Docking Analysis and Study of Permeability

Keap1 (PDB ID: 4ZY3) was subjected to the docking study. Protein and active site coordinates were directly extracted from PDB ID: 4ZY3 from the Protein Data Bank website https://www.rcsb.org/ (accessed on 14 October 2022). UCSF Quimera 1.15 Software (RBVI, San Francisco, CA, USA) was employed to delete unwanted residues from protein. The Autodock Tools 1.5.6 software (Scripps Research, San Diego, CA, USA) was used to analyze the prepared protein and ligands for molecular docking. The molecular docking was achieved with AutoDock Vina software, https://vina.scripps.edu/ (Scripps Research, San Diego, CA, USA), as previously described [64].

The in silico constructed ligands **1**, **1a**, and **1b** were subjected to the Monte Carlo search protocol considering a 0–5 kcal/mol energy gap using the MMFF94 force field, executed in the Spartan’04 program. The resulting conformers as the global minimum were geometry-optimized by the DFT B3LYP/DGDZVP level of theory in the Gaussian 16 software (Wallingford, CT, USA). Geometry-optimized conformers were submitted to the docking study (Appendix A).

Additionally, a study of parameters (iLOGP, XLOGP3, WLOGP, MLOGP, SILICOS-IT) that allow the prediction of the permeability of a pure compound through a membrane was carried out to estimate the theoretical lipophilicity for pattern phenolic **1** and chrysin glucoside derivative **1a** by using the SwissADME web tool, since **1a** showed a better bioactive profile.

### 4.11. Statistical Analysis

Data generated from the mentioned assays are expressed as arithmetic means ± standard error of the mean (SEM) and were statistically analyzed with the GraphPad Prism^®^ Version 6.00 software (GraphPad Software, Inc., San Diego, CA, USA). The Kolmogorov–Smirnov test was used to confirm the normality of the data. The Student’s *t* test was used to compare the two control groups (Control and LPS). One-way ANOVA was conducted to determine statistical significance between several groups, followed by Tukey’s Multiple Comparison test for parametric data. A *p* value was considered significant if it was less than 0.05. Additionally, the software G-Power version 3.1.9.7. was used to calculate the minimal sample size for each parameter tested with 80% statistical power.

## 5. Conclusions

In conclusion, this study reports for the first time that chrysin-8-*C*-glucoside and two new derivatives suppressed oxidative stress and the inflammatory response caused by LPS in human THP-1 macrophages by reducing intracellular ROS levels, as well as TNF-α, IL-1β, and COX-2 production. The mechanisms involved in these effects are associated, at least in part, with the activation of the Keap-1/Nrf2/HO-1 pathway. The findings of the docking assay support the use of these phenolic compounds as Nrf2 activators, as they appear to compete to bind to the natural Nrf2 inhibitor, Keap1, allowing the activation of the transcription factor Nrf2 and, therefore, the expression of its target gene HO-1, as shown in the mechanism of action proposed in Figure 7. This is an in vitro study with promising results showing new Nrf2 activators, with compound **1a** presenting the highest lipophilicity and, thus, a better biological activity profile. Thus, **1a** could be used for further studies to elucidate additional action mechanisms and its quantitative structure–permeability relationship. However, this study presents some limitations, such as the lack of in vivo models to test the potential bioactivity of the phenolics; therefore, further research is needed to explore the metabolism, bioavailability, and potential toxicity of the compounds in vivo. In this way, the research findings could be translated into clinical practice since these bioactive phenolics have the potential to be used as nutraceuticals in oxidative-stress-related inflammatory diseases.

## Figures and Tables

**Figure 1 pharmaceuticals-17-01388-f001:**
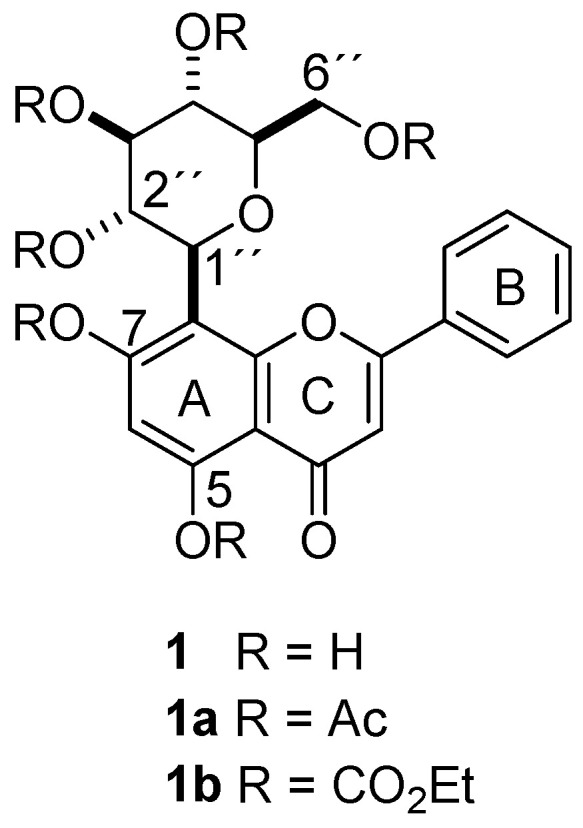
Formulas of chrysin-8-*C*-glucoside (**1**) and the respective hexa-acetate derivative (**1a**) and the hexa-ethyl carbonate derivative (**1b**).

**Figure 2 pharmaceuticals-17-01388-f002:**
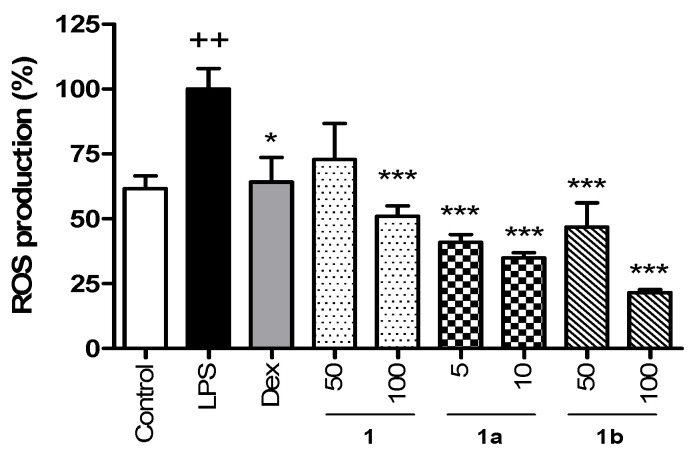
Effect of phenolics **1**, **1a**, and **1b** on the intracellular ROS production in THP-1 macrophages stimulated by LPS. These cells were treated with concentrations of 50 and 100 µM for compounds **1** and **1b** and 5 and 10 µM for compound **1a** for 1 h and then stimulated with 1 µg/mL LPS for 24 h. The positive reference drug dexamethasone (Dex) was used at 1 µM. Phenolics reduced the intracellular ROS production in a concentration-dependent manner even below Control levels at the higher tested concentrations. Values are representative of four independent experiments (*n* = 4). Data are arithmetic means ± SEM plotted by vertical bars. Mean value was significantly different vs. Control (++ *p* < 0.01; Student’s *t* test). Arithmetic mean values were significantly different in comparison with LPS (* *p* < 0.05, *** *p* < 0.001; Kruskal–Wallis test followed by Dunn’s Multiple Comparison test.

**Figure 3 pharmaceuticals-17-01388-f003:**
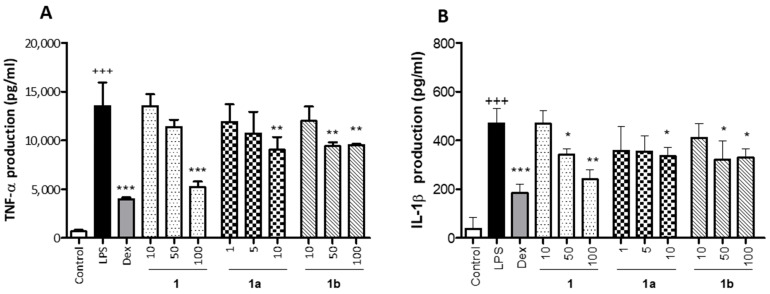
Effect of phenolics **1**, **1a,** and **1b** on LPS-induced TNF-α (**A**) and IL-1β (**B**) production in THP-1 macrophages. These cells were treated with the concentrations of 10, 50 and 100 µM for the compounds **1** and **1b** and 1, 5, and 10 µM for the compound **1a**, for 1 h and then stimulated with 1 µg/mL LPS for 24 h. Cytokines were quantified in cellular supernatants by using ELISA assay. The positive reference drug dexamethasone (Dex) was used at 1 μM. Values are arithmetic means ± SEM from six independent experiments (*n* = 6). Data are arithmetic means ± SEM plotted by vertical bars. Mean value was significantly different vs. Control (+++ *p* < 0.001; Student’s *t* test). Arithmetic mean values were significantly different in comparison with LPS group (* *p* < 0.05, ** *p* < 0.01, *** *p* < 0.001; one-way ANOVA followed by Bonferroni’s Multiple Comparison test).

**Figure 4 pharmaceuticals-17-01388-f004:**
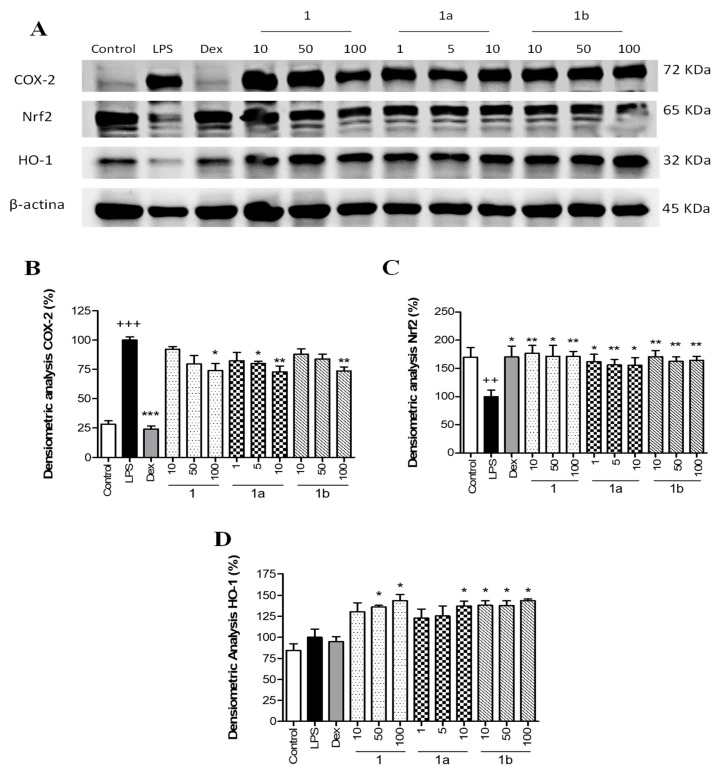
Effect of phenolics **1**, **1a**, and **1b** on the pro-inflammatory *COX-2* expression and the *Nrf2/HO-1* signaling pathway in THP-1 macrophages stimulated by LPS. These cells were treated with concentrations of 10, 50, and 100 µM for compounds **1** and **1b** and 1, 5, and 10 µM for compound **1a** for 1 h and then stimulated with 1 µg/mL LPS for 24 h. The positive reference drug dexamethasone (Dex) was used at 1 μM. (**A**) Representative Western blotting analysis of COX-2, Nrf2, and HO-1 proteins from the same membrane. Densitometric analysis of COX-2 (**B**), Nrf2 (**C**), and HO-1 (**D**) carrying out normalization to β-actin. Results are representative of four independent experiments (*n* = 4). Values are arithmetic means ± SEM plotted by vertical bars. Arithmetic mean value was significantly different vs. Control (+++ *p* < 0.001, ++ *p* < 0.01; Student’s *t* test). Arithmetic mean values were significantly different in comparison with the LPS group (* *p* < 0.05; ** *p* < 0.01; *** *p* < 0.001; Kruskal–Wallis test followed by Dunn’s Multiple Comparison test).

**Figure 5 pharmaceuticals-17-01388-f005:**
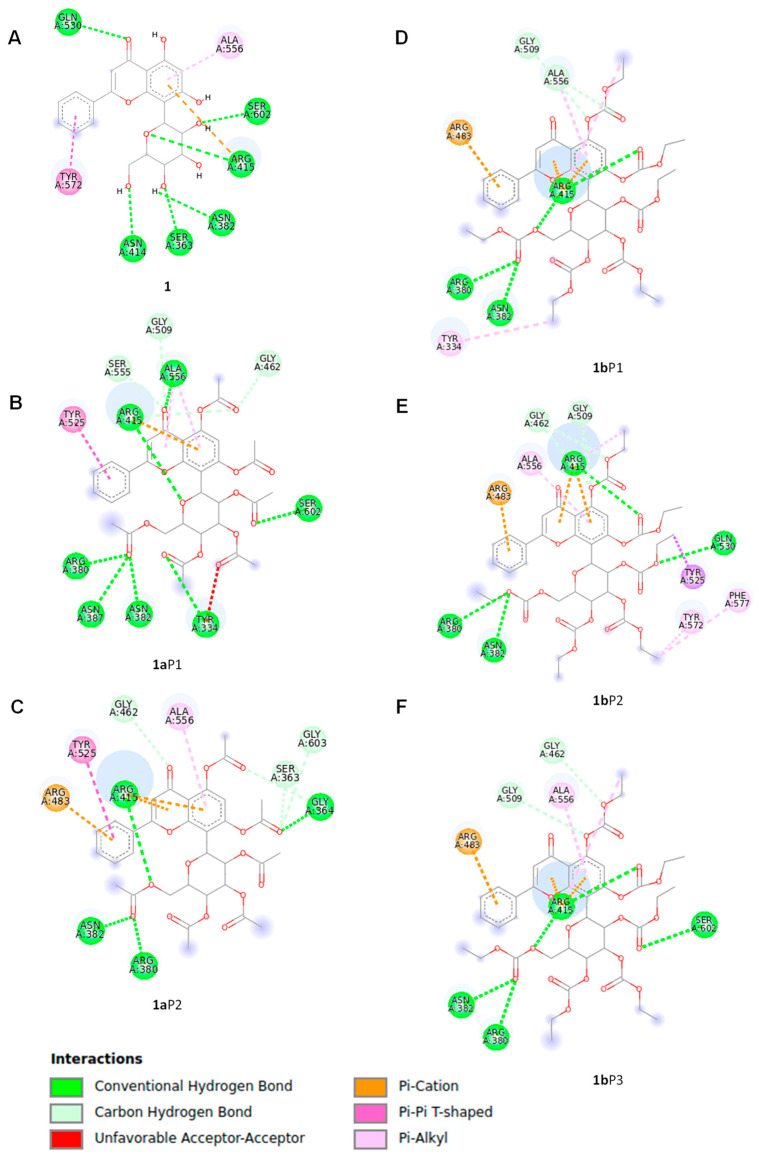
Molecular interactions of compounds **1**, **1a**, and **1b** with Keap1 calculated by docking protocols. The protein data (PDB ID: 4ZY3) were extracted from the Protein Data Bank website (https://www.rcsb.org/). The docking protocol included the UCSF Quimera 1.15, Autodock Tools 1.5.6, and AutoDock Vina software. The molecular models of ligands **1**, **1a**, and **1b** involved in the docking study were geometry-optimized global minimum conformers, which were found by the Monte Carlo search protocol in the MMFF94 force field, as executed in the Spartan’04 program, followed by geometry optimization by the DFT B3LYP/DGDZVP level of theory in the Gaussian 16 software. (**A**) One pose was determined for glucoside flavonoid **1**, while two poses were found for derivative **1a** represented by (**B**) **1a**P1 and (**C**) **1a**P2. Compound **1b** showed three poses including (**D**) **1b**P1, (**E**) **1b**P2, and (**F**) **1b**P3.

**Figure 6 pharmaceuticals-17-01388-f006:**
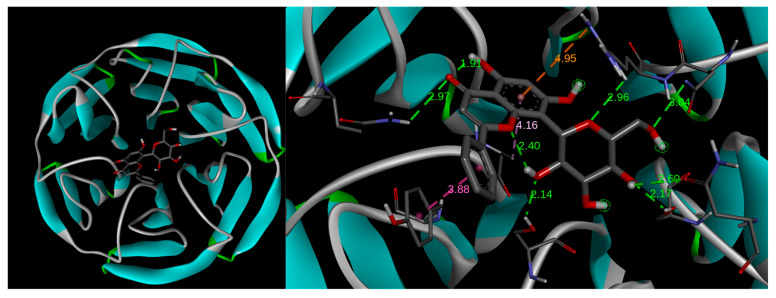
Docking models of compound **1**. The **left** image shows the Kelch domain of Keap1, which is a segment of the protein constituted by the 6-bladed *β*-propeller structure interacting with compound **1**. The **right** image shows the rings A and B from the flavonoid skeleton providing recurrent molecular interactions with Ala556, Arg415, and Asn382 amino acids from Keap1.

**Figure 7 pharmaceuticals-17-01388-f007:**
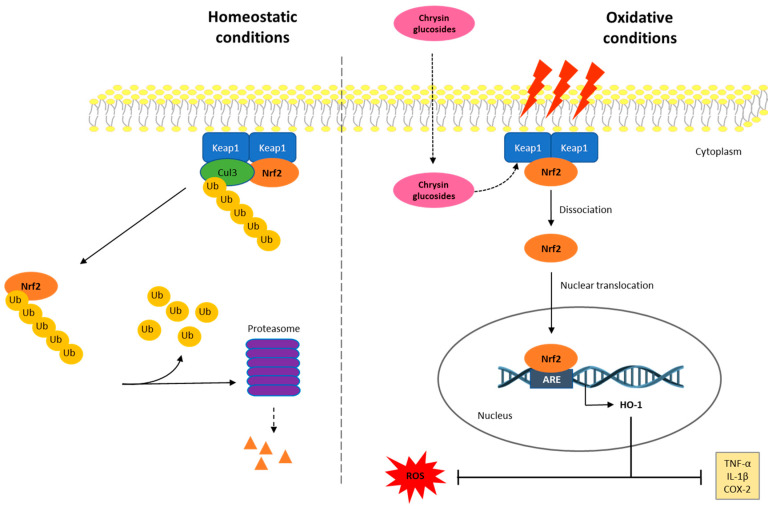
Proposed diagram for the anti-inflammatory activity of phenolics **1**, **1a**, and **1b** through the interaction with the Keap1/Nrf2/HO-1 signaling pathway. In homeostatic conditions, the Keap1–Nrf2 system is placed joined to the cell’s internal hemimembrane. In these conditions, inactive Nrf2 is occasionally ubiquitinated and is degraded in the proteosome. In oxidative conditions, the intracellular ROS production and the inflammatory process are triggered, and the Nrf2 signaling pathway may be activated to control ROS-induced damage. Then, Nrf2 is dissociated from the Keap1 dimer and migrates into the nucleus to provide the transcription of its target gene, HO-1. This fact leads to the down-regulation of ROS production and inflammatory processes by inhibiting TNF-α, IL-1β, and COX-2 levels. In oxidative stress conditions, phenolics **1**, **1a**, and **1b** may be incorporated into the cell and act as Keap1 ligands, causing Nrf2 not to bind to or dissociate from Keap1, leading to further Nrf2 activation.

**Figure 8 pharmaceuticals-17-01388-f008:**
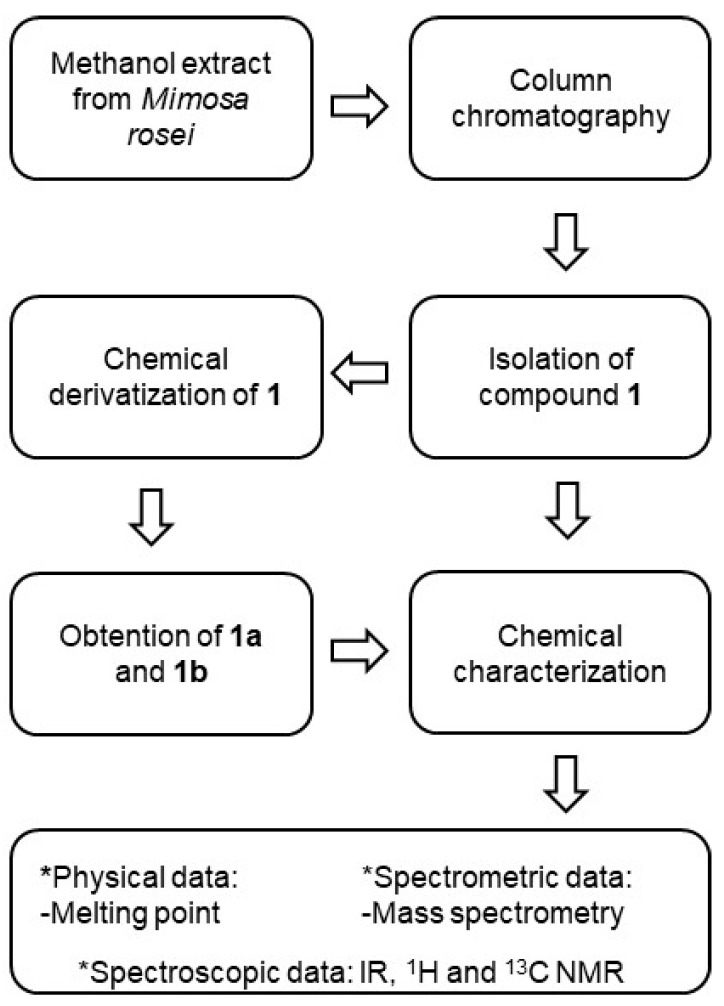
Methodology for the obtention and chemical characterization of **1**, **1a**, and **1b**.

**Table 1 pharmaceuticals-17-01388-t001:** Binding affinity and molecular interactions of ligands **1**, **1a**, and **1b** with Keap1 (PDBID: 4ZY3).

Ligand	Binding Affinity ΔG, (kcal/mol)	Aminoacids of Keap-1 Receptor Forming Intermolecular Interactions	Types of Interaction
**1**	−8.9	Ser363, Asn382, Asn414, Arg415, Gln530, Ser602	Conventional Hydrogen Bond
		Arg415	Pi-Cation
		Tyr572	Pi-Pi Stacked
		Ala556	Pi-Alkyl
**1a**P1	−8.2	Tyr334, Arg380, Asn382, Asn 387, Arg415, Ala556, Ser602	Conventional Hydrogen Bond
		Arg415, Gly462, Gly509, Ser555	Carbon Hydrogen Bond
		Arg415	Pi-Cation
		Tyr525	Pi-Pi T-shaped
		Ala556	Pi-Alkyl
		Tyr334	Unfavorable Acceptor-Acceptor
**1a**P2		Gly364, Arg380 Asn382, Arg415	Conventional Hydrogen Bond
		Ser363, gly,364, gly462, gly603	Carbon Hydrogen Bond
		Arg415, Arg483	Pi-Cation
		Tyr525	Pi-Pi Stacked
		Ala556	Pi-Alkyl
**1b**P1	−7.9	Arg380, Asn382, Arg415	Conventional Hydrogen Bond
		Gly509, Ala556	Carbon Hydrogen Bond
		Arg415, Arg483	Pi-Cation
		Tyr334, Arg415	Alkyl
		Ala556	Pi-Alkyl
**1b**P2		Arg380, Asn382, Arg415, Gln530	Conventional Hydrogen Bond
		Gly462, Gly509	Carbon Hydrogen Bond
		Arg415, Arg483	Pi-Cation
		Tyr525	Pi-Sigma
		Arg415, Tyr572	Alkyl
		Ala556, Phe577	Pi-Alkyl
**1b**P3		Arg380, Asn382, Arg415, Ser602	Conventional Hydrogen Bond
		Gly462, Gly509	Carbon Hydrogen Bond
		Arg415, Arg483	Pi-Cation
		Arg415	Alkyl
		Ala556	Pi-Alkyl
Native ligand	−9.3	Ser508, Ser602	Conventional Hydrogen Bond
		Arg415	Pi-Cation
		Ala556	Pi-Sigma
		Tyr334, Tyr525, Phe577	Pi-Pi
		Tyr572	Pi-Alkyl

## Data Availability

The original contributions presented in the study are included in the article/Appendix A, further inquiries can be directed to the corresponding author.

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
