# Peer review of "Synthesis and Bioevaluation of New Stable Derivatives of Chrysin-8-C-Glucoside That Modulate the Antioxidant Keap1/Nrf2/HO-1 Pathway in Human Macrophages"

_pharmaceuticals, 2024, doi:10.3390/ph17101388_

Round 1
Reviewer 1 Report
Comments and Suggestions for Authors
The introduction does a good job of describing the inflammatory response, the role of ROS, and flavenoids, but does not explain the Keap1/Nrf2/HO-1 pathway, which is necessary to understand the mechanism of the active compounds. Further, the experimental and theoretical methods the authors employed should be mentioned in the introduction before discussing the results.
The methods are well-described and cover both the docking simulations and the experimental characterization.
The figures are clear and help to explain the results. However, the docking findings need additional explanation of why this target was chosen, what is the kelch domain, what protocol was employed, etc.
Figure 7 should be moved to the discussion section rather than the conclusion. Also the reference to Figure 7 in the text is currently mislabelled as Figure 6.
The English language also requires some work, as shown in the section below.
Overall, I would recommend this article for publication after minor revisions.
Comments on the Quality of English LanguageThe English in this article is mostly well-written. However, there are a number of spelling and typographical errors that should be corrected before publication. A few examples are given below, but the entire manuscript should be checked thoroughly.
Page 2, line 53: "as for example antioxidant, anti-inflammatory, anti-cancer and anti-microbial" should be "for example as antioxidant, anti-inflammatory, anticancer and antimicrobial compounds"
Page 2, line 58: "thermodinamics" should be "thermodynamics"
Page 2, line 80: "chrysin as nutraceutical in" should be "chrysin as a nutraceutical in"
Page 3, line 119: "Its whose HRMS (ESI+) showed" should be "The HRMS of this derivative (ESI+) showed"
Page 9, line 287: "after the chromatography process" should be "after purification by chromatography"
Reviewer 2 Report
Comments and Suggestions for Authors
- Streamline the introduction and discussion to focus on the most critical findings and their relevance in the current scientific landscape.
- Discuss the limitations and potential clinical applications more thoroughly, including challenges in translating in vitro findings to in vivo or clinical settings.
-Add more detailed figure captions and explanations, especially for complex docking models, to improve accessibility for readers unfamiliar with molecular docking techniques.
Introduction:
There is a significant amount of background information on flavonoids in general, which could be streamlined to focus more on previous work specifically on chrysin derivatives. Furthermore, while the introduction touches on inflammation and oxidative stress, it could briefly expand on why macrophages were chosen as the model system.
Methodolgy:
The methodology is quite dense, particularly the chemical characterization section, and might overwhelm readers unfamiliar with advanced chemistry. A flowchart or schematic of the synthetic pathways and experimental workflow could help. Additionally, a brief mention of statistical power and sample sizes would strengthen the validity of the findings.
Results:
While the data on antioxidant activity and cytokine reduction are presented well, the discussion on how these findings compare to existing literature on other derivatives or natural antioxidants is lacking. Furthermore, some results (like the docking study) could benefit from more detailed interpretation, such as how specific amino acid interactions might affect binding affinity and activity.
Discussion:
The discussion could be expanded to consider potential clinical applications and challenges in drug development, such as bioavailability in vivo, dosage concerns, or side effects. Additionally, the study does not address limitations, such as the lack of in vivo testing or potential toxicity of the derivatives in higher concentrations.
Comments on the Quality of English Language
Minor editing of English language required.
Round 2
Reviewer 2 Report
Comments and Suggestions for Authors
Accept.
Comments on the Quality of English LanguageMinor editing of English language required.